# Perceptions and acceptability of microsampling in children and young people: a single-centre survey

Andrew J Chetwynd  ,[1] Julien Marro,[1] Laura Whitty,[1] Sammy Ainsworth,[2] Jennifer Preston,[2] Alan Salama,[3,4] Louise Oni[1,5]

[1]Department of Women's and Children's Health, University of Liverpool, Liverpool, UK
[2]NIHR Alder Hey Clinical Research Facility, Alder Hey Children's NHS Foundation Trust, Liverpool, UK
[3]Centre for Nephrology, University College London, London, UK
[4]Department of Nephrology, Royal Free NHS Hospital, London, UK
[5]Department of Paediatric Nephrology, Alder Hey Children's NHS Foundation Trust Hospital, Liverpool, UK

**Correspondence to**
Dr Louise Oni; louise.oni@liverpool.ac.uk

## ABSTRACT

**Background** The use of at home microsampling devices, such as dried aliquots of blood and urine, for scientific research has expanded in recent years. These devices collect small volumes of biofluids which air dry and can be posted to central laboratories. In general, they are cheaper and more convenient, saving patients travel time and freeing up staff. In adult populations, this sampling method is well perceived, with >90% of samples being of adequate size and quality for scientific research. However, little literature on microsampling in paediatric populations exists. The aim of this study was to explore the perceptions and acceptability of microsampling in children and young people.

**Methods** Online patient and public involvement was obtained by attending the Generation R Young Person's Advisory Group Liverpool, to explore the perceptions of at home microsampling. During the session, the rationale and use of microsampling was demonstrated. The participants provided informal feedback during the interactive session and completed a short online questionnaire.

**Results** A total of 13 children and young people attended the event and they were aged 10–17 years, of these 10 responded to an online questionnaire. The general feedback on microsampling was positive with 80% of respondents indicating they would be willing to participate in at home microsampling studies. Furthermore, 100% respondents reported being willing to provide both biofluids on a monthly basis and 40% would be willing to provide dried urine samples weekly.

**Conclusions** Children and young people are supportive of at home microsampling for research purposes and this offers the possibility of widening participation to research.

## BACKGROUND

The collection of biofluid samples from patients for basic science and clinical research typically requires a patient to attend an appointment with a medical practitioner, particularly in the case of providing blood samples.[1 2] Even in the case of a urine sample, a visit to a hospital or research facility is frequently required, incurring further costs and time commitments to families. Recently, there has been a growing interest in micro sampling such as dried blood and urine spots, collected at home and posted back to the

### WHAT IS ALREADY KNOWN ON THIS TOPIC

⇒ New born screening using dried blood spots have been used for decades and the use of dried blood spots is adults is growing in popularity.
⇒ The use of dried blood spots in adults has been shown to be applicable for many questions such as adherence to treatment regimen.
⇒ There is evidence from adults that they prefer this method of sample collection due to its less invasive nature particularly when economic and time costs are accounted for.

### WHAT THIS STUDY ADDS

⇒ The small group of children and young persons in this study were happy with the collection of dried blood spots. This opens the possibility for more children and young persons to contribute to medical research whom may have been excluded previously and may ease longitudinal sampling.

### HOW THIS STUDY MIGHT AFFECT RESEARCH, PRACTICE OR POLICY

⇒ This may make decisions regarding ethical approvals easier given that this work gives this population a voice.
⇒ These findings may help accelerate patient recruitment and sampling to help resolve the gap in research between paediatric and adult diseases.

research centre for analysis.[2] The use of dried biofluids is cheaper[1 3 4] and an array of studies show the samples are stable over a few days, which encompass the necessary postal time.[5] In addition, as a dried biofluid, the biological risk associated with the liquid form is significantly reduced.[2 6] Since the COVID-19 pandemic, home testing of nasal/throat swabs have become common practice.

To date, there has been a growing number of studies evaluating home sampling and postage delivery spanning personalised nutrition,[7] genome testing,[8] disease monitoring and preclinical research.[4 9 10] In addition to the scientific questions at the heart of these studies, a number have looked into patients' perception and sample quality

of microsampling. The overarching consensus is that participants are happy to provide blood, urine and saliva samples using micro sampling devices with <10% of patients preferring traditional clinical visits. The reasons for the latter being due to confusion over the sample collection, fear of pain with the finger prick blood devices or not feeling adequately trained in the sampling technique.[11] Furthermore, these studies report high levels of compliance with >90% of expected samples being taken and successfully arriving for analysis, even in the case of longitudinal sampling.[3 12]

These studies highlight the promise of at home microsampling to increase recruitment of patient populations that are not encompassed within a research-active hospital or university region and facilitate larger cohort recruitment with longitudinal sampling studies.[4]

There are little data evaluating at home microsampling devices in paediatric cohorts, despite paediatric healthcare professionals being familiar with national newborn screening programmes. Using this technique in children and young people may also be dependent on the willingness of research ethics committees to grant ethical approval for at-home microsampling testing in a vulnerable population and thus there is a threat that children may be excluded from this research area. The aim of this work is to explore the perceptions of and willingness of children and young people to partake in research focused on at-home microsampling of biofluids.

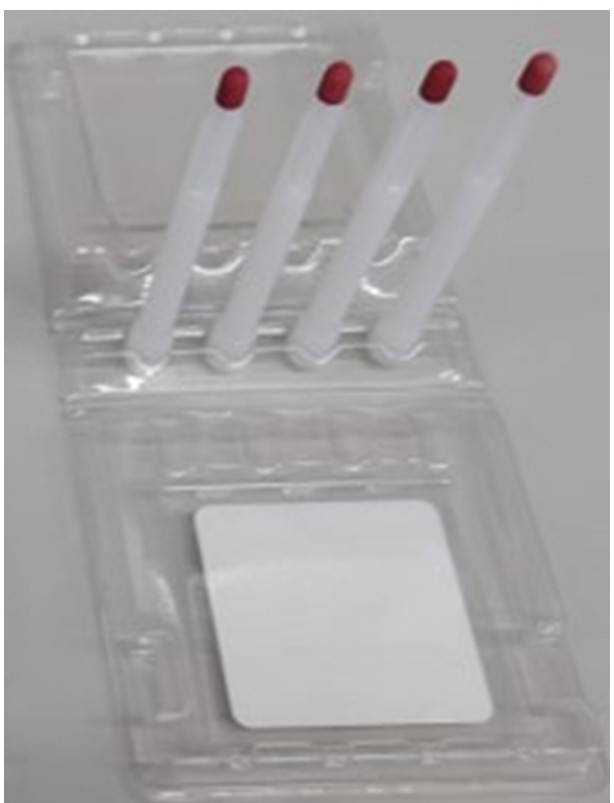

**Figure 1** VAMS devices for the collection dried blood spots generated from 30 µL of capillary blood. VAMS, volumetric absorptive microsampling.

## METHODS

Researchers (LO (female), AC (male)) applied to attend one of the GenR Young Person's Advisory Group (YPAG) Liverpool meetings. The Liverpool YPAG was set up in 2006 and is open to young people between the ages of 8–23 years. The group currently has 28 members aged between 10 and 22 years old. Members either have experience of taking part in health research, have experience of living with a condition or disability, or have a general interest in learning about medicine/research/science. There were no inclusion or exclusion requirements for the study other than being a member of GenR YPAG group, any member was free to participate.

Prior to the workshop, meetings were held with the group facilitator (SA) to plan the session as part of a regular GenR YPAG meeting and to support the purpose of the activity including expected outputs. The dissemination of a sample of the pack were also arranged. It was decided to send packs to a small group of members who would demonstrate to the rest of the group during the meeting. In total six mock sampling packs were sent to YPAG members to facilitate a live demonstration. In the mock sampling pack, volumetric absorptive microsampling (VAMS, Neoteryx, California, USA) devices were provided with two aliquots of food colouring to mimic blood and urine. Two aliquots of water were also provided to allow a demonstration of how to recover the dried simulated biofluids.

The online meeting was conducted over 45 min using Zoom video conferencing software (Zoom video communications, California, USA), the session was not recorded. Researchers (LO and JA) presented current recruitment and sample collection practices and the importance of these for understanding inflammatory renal disease. The session subsequently involved a short video and a live interactive demonstration (AC) to illustrate the sample collection process using the mock sampling kits. Mock samples (VAMS and food colouring) were left to dry for 20 min during the discussion phase prior to illustrating how to recover the dried sample using the provided water.

During the presentation, a short video was shown demonstrating the use of a finger prick device for the generation of a capillary whole blood sample and its collection with a VAMS device (figure 1). The group then discussed with the whole group for 35 min the positive and negative points to sample collection and finally a multiple-choice questionnaire (SurveyMonkey Audience www.surveymonkey.com/mp/audience) (online supplemental information 1) was circulated to assess their perception and willingness for children and young people to partake in studies using microsampling devices. During the meeting written minutes were recorded by two individuals (LO and JM). The responses to the multiple-choice questionnaire was tallied and graphs produced (AC) using GraphPad Prism V.8.1.1 (GraphPad Software,

California, USA). The trends in perception were then garnered from this compiled data.

## Patient/public involvement

This work collected the views and perceptions of children and young persons from the public to inform and improve future at home dried biofluid microsampling clinical research studies.

## RESULTS
## Participants

In total, 13 young people (46% of members, no parents present) attended the YPAG meeting via Zoom from home in April 2022. Only one young person of six received the mock pack and was able to attend due to educational and personal commitments. They confirmed that the mock sampling pack was able to fit through their letter box and had been received intact. Due to the small group data saturation was not reached.

## Findings from informal discussion

When asked for first impressions on microsampling, the feedback was positive with comments on the method being 'quick', 'efficient', 'simple', 'helpful', 'fast', 'great' and 'easy' being made. The participants were asked for any advantages of using these microsampling collection kits which followed a similar trend with responses of 'Don't need a lot of volume of urine/blood', 'simple to use and also fewer travelling costs' and 'only takes ten minutes rather than a day to go to hospital'.

Conversely, a number of concerns were raised which can either be readily resolved with added information or require additional work prior to rolling out microsampling (table 1). The participant doing the hands-on demonstration mentioned that initially the procedure seemed 'complicated' but ended up being 'easy and simple to operate'. This raised a further question 'will there be an instruction guide or video?', the initial idea behind this form of sample collection was to provide a manual with a step by step process. However, the group suggested using a QR code with an embedded link to a guidance video or online manual.

An issue was raised about whether the results from these samples are comparable to standard onsite venous blood and urine sample collection. The issue of still having to collect a urine sample in a container prior to taking the microsample was also raised highlighting the need to investigate an appropriate container in the future. However, it was emphasised that collecting a urine sample at home was more convenient and more private compared with a hospital visit. The supply of finger prick devices in the packs to be shipped out was also questioned and it was confirmed that plans would be to include these with spares in case one did not work or multiple pricks were required. The final concern raised was the environmental impact of such testing, demonstrating the

**Table 1** Concerns raised by the group and the impact they may have on future studies with potential solutions discussed

| Concerns raised | Potential issues this would have | Possible solutions |
|---|---|---|
| Complexity of sample collection/requirement for instruction | Poor quality sample<br>Unnecessary pain if blood taken incorrectly<br>Stress of participant | Develop manuals and videos in accessible language<br>Use of further PPI events to evaluate instructional manuals and videos.<br>Incorporation of QR codes to link to videos and manuals |
| Lack of data on how microsampling compared with traditional techniques | Lower quality data being collected<br>Not enough samples<br>Sample degradation | Prior to each study a method development phase is required to optimise sample recovery, limits of detection, sample stability. |
| Urine collection in a separate container | Inconvenient<br>Large volume to dispose of without spilling | Focus group highlighted that this is still preferable to in person sampling in clinic. However, it's something to consider for researchers to make the process as easy as possible. |
| Will the kits posted home contain the finger prick devices? | It could limit the recruitment of patients if they think they need to provide materials for sample collection | Full description of the process and kit contents in information packs for participants.<br>Ensuring adequate information in manuals and instructional videos. |
| Environmental impact of the microsampling kits | If perceived to be greater than traditional approaches it may deter participants.<br>It's important that clinical researchers consider the impact on global pollution burden as this contributes to poor health outcomes. | At home sample collection reduces the need for transport to a research site.<br>Less plastics are required compared with traditional methods.<br>Researchers should assess a number of microsampling devices to minimise the use of single use plastics. |

PPI, patient and public involvement.

environmental concern of patients alongside medical concerns.

In terms of the ages that the group thought would be appropriate for microsampling, there was no obvious age cut-off. They suggested that it would need to be done on a case by case basis and felt that if young people under 16 years old were able to understand and consent then they should be allowed to participate and that some children and young people may actually prefer this method of sampling compared with traditional venous sample collection.

## Findings from questionnaire

The YPAG were asked to answer a multiple choice questionnaire (online supplemental information) and online supplemental table 1) to help evaluate their perceptions. Initial questions revolved around the use of microsampling devices for clinical research in paediatric cohorts. In total, 10/13 participants (77%) provided feedback to the questionnaire and of those, 9/10 felt that microsampling was preferential compared with having to visit the general practitioner (GP) or a hospital site for blood drawing and 1/10 respondent was unsure. Furthermore, 8/10 of respondents thought that this approach would increase the likelihood of patients wishing to participate in clinical research while 2/10 were unsure.

A key question with this type of microsampling in this population is to understand which biofluids they feel comfortable with providing and this may provide information for ethical committees to make an informed decision on their behalf. Perhaps surprisingly, a great proportion were willing to provide a blood sample (9/10) compared with a urine sample (8/10), despite the use of a finger prick test to obtain the capillary blood sample. In the questions regarding both the perception of the microsampling devices and the biofluids, between 1/10 and 2/10 of respondents replied with 'unsure'.

Finally, the frequency of sampling is significant for the recruitment of patients over a defined time frame with periodic sampling providing a great opportunity for longitudinal data collection as the disease changes over time and treatment takes effect. Perhaps unsurprisingly, there was a trend for blood to be collected on a less regular basis, potentially due to apprehension regarding the finger prick testing. The frequency of urine testing was less clear cut with 4/10 suggesting that weekly collection or monthly collection would be acceptable (figure 2).

## DISCUSSION
## Initial discussion

The initial positive responses to the use of at home microsampling and return by mail reflect the findings in adult cohorts about the use of such devices that they would be quicker and easier to use and cut down on time and economic costs associated with a hospital or GP visit.[12]

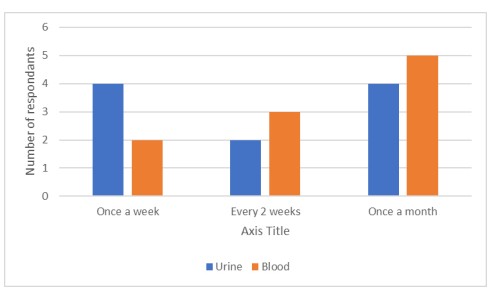

**Figure 2** The Liverpool YPAG opinions on the hypothetical acceptable frequency of microsampling depending on sample type. YPAG, Young Person's Advisory Group.

Importantly, a number of potential weaknesses were also highlighted by the group. The issues raised over the complexity and level of instruction for the at home sampling is a vital aspect of improving inclusivity enabling children, young people and parents to make informed choices and safely and reliably carry out the sample collection procedure. In order to be as inclusive as possible with this style of sampling, there is a clear need for providing instructions accessible for all to ensure safe and reliable at home microsample collection. A further patient and public involvement (PPI) event would be beneficial to evaluate the manual or videos in order to optimise these for this audience. These issues are also reflected in adult cohorts where a number of participants requested further guidance for microsampling and when it was made available participation rates were increased.[3 11]

Concerns regarding quality of data that can be ascertained from microsampling compared with more traditional approaches is an understandable concern as patients may be asked to provide samples more frequently than typical during their clinical follow-up so it is important that researchers can reassure patients that the samples they are providing are being put to good use. In an array of studies in adult populations >90% of returned samples were of adequate volume and quality for research purposes, suggesting that this approach would be appropriate.[3 11–13] However, validation and stability for the specific analytes of interest will be required for all studies using this form of sample collection, particularly given that reference ranges are typically provided for serum or plasma rather than whole blood.[11 14] Once this has been validated, it would possibly be useful to include this information and additional information of previously published studies to show the applicability of the sampling technique to potential patients interested in providing dried biofluid samples.

The group highlighted a concern of the environmental impact of the plastics usage with this microsampling technique. Investigators will need to consider the type of plastic and that by removing the requirement to travel to a research centre and the reduced sample processing in the laboratory[1 3] may balance the environmental impact compared with conventional sampling. Furthermore, the suggestion of QR codes for access to videos and manuals could reduce paper work and the weight of the packaging

however this may disadvantage individuals without digital access to take advantage of this.

## PPI questionnaire

The reported willingness to participate in this style of study is much higher than the response rate observed in an adult epidemiological cohort and may reflect that this was a small group or the difference between a theoretical study and one that is actually taking place.[13] This is key as this population is already under represented in the literature potentially due to the low numbers of specialist paediatric research facilities, spare biofluid volume for clinical research or reticence to provide ethical consent for clinical research on this population. All of which combine to limit the amount of research on paediatric diseases potentially leaving this population behind in terms of mechanistic insights, discovery of potential drug targets and the development of new drugs.

The slight preference for blood sampling over urine is particularly interesting because urine is often stated as a preferred biofluid for research due to it being less invasive and more plentiful. It is possible that this is down to having to provide a sample into a large pot and then taking a sample or embarrassment around providing a urine sample. In addition, the disposal of a container full of urine once the sample has been taken may also put participants off.

The YPAG group responded that they would be open to longitudinal collection of biofluids, though there was no consensus regarding the frequency of sample collection. The results suggested that aiming monthly sampling may be acceptable however further work using specific cohorts may be required.

## Study limitations

While the findings from this work are promising for at home microsampling in young persons and children there are number of important limitations to consider. This study only used a very small number of individuals (n=13) spanning a relatively large age range, additionally only 10 responded to the questionnaire. These individuals also have an interest in medical research and are members of the Liverpool GenR younger persons advisory group. As such these individuals are potentially more inclined to find this approach favourable. It is also very difficult to simulate the experience of a finger prick test using a video and individuals may change their perceptions of this approach following their first experience. However, this effect may be less evident for the use of dried urine spots. This work was also complicated by the fact only one participant received the at home demonstration kit to evaluate for the session. This work offers some insight into the views and thoughts on this topic but larger follow-up studies before roll out of at home microsampling will be vital to evaluate the views of young adults and children on this process.

## CONCLUSIONS

While the group size is small and there are biases due to the intellect and interest in clinical research held by members possibly skewing the findings, we are able to report that children and young people may be willing to provide samples in this manner. We also gained insight on the frequency and additional concerns that need to be addressed for longitudinal research studies aimed at elucidating mechanistic insights into paediatric diseases. These findings may help inform ethics boards for this style of research.

**Acknowledgements** We would like to thank the Liverpool GenR YPAG for allowing us to present our proposal and for their critical and thoughtful feedback to this work.

**Contributors** LO is the senior author and the author responsible for the overall content as guarantor. LO and AS obtained funding and led the concepts behind aspects of the PPI event. LO, AC, JM, LW and SA produced the demonstration kits and posted out to participants. AC and LO wrote the initial draft of the manuscript. JM kept minutes of the PPI event. LW, JP and SA arranged and set up the PPI event and SA led the overall PPI session with LO and AC presenting the sampling technique and providing the research background. JP manages the PPIE programme and has contributed to the final manuscript. All authors were responsible for critical revision of the manuscript and approved the final version.

**Funding** This research was funded by the Funding Autoimmune Research (F.A.I.R.) Charity (Registered UK Charity; number: 1176388) and Kidney Research UK Meditech academy programme The Liverpool GenR YPAG is funded by the NIHR Alder Hey Children's NHS Foundation Trust Clinical Research Facility (CRF). This work was carried out with in the Experimental Arthritis Treatment centre for children funded by Versus Arthritis.

**Competing interests** None declared.

**Patient and public involvement** Patients and/or the public were involved in the design, or conduct, or reporting, or dissemination plans of this research. Refer to the Methods section for further details.

**Patient consent for publication** Not applicable.

**Ethics approval** This study involves human participants but all work with GenR Liverpool was carried out after detailed meetings and approved via a predefined Researcher Agreement with GenR members and authorised by the PPIE Manager (2022). Participants gave informed consent to participate in the study before taking part.

**Provenance and peer review** Not commissioned; externally peer reviewed.

**Data availability statement** All data relevant to the study are included in the article or uploaded as online supplemental information.

**ORCID iD**
Andrew J Chetwynd http://orcid.org/0000-0001-6648-6881

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
