## [Reviewer comments · BMJ Paediatrics Open]

ARTICLE DETAILS

TITLE (PROVISIONAL)	Perceptions and acceptability of micro-sampling in children and young people; a single centre survey
AUTHORS	Chetwynd, Andrew Marro, Julien Whitty, Laura Ainsworth, Jenny Preston, Jennifer Salama, Alan Oni, Louise

VERSION 1 – REVIEW

REVIEWER	Reviewer name: Dr. Helen M Sammons Institution and Country: North Devon District Hospital, Paediatrics, United Kingdom of Great Britain and Northern Ireland Competing interests: None
REVIEW RETURNED	30-Nov-2022

GENERAL COMMENTS	A well written piece that adds to our knowledge. A small descriptive group discussion that is documented to show children & young people's views on microsampling. The sample size is biased & small (13 attended & 10 completed a non validated survey). The paper acknowledges this noting it is also an interested population who have a health care/research interest. This could be brought out further in the abstract. It should also be highlighted further in the limitations the fact it was an online session with only one participants having hands on access to the kit. It however gives the first evidence of this groups thoughts. It will hopefully be the start of further consumer groups to look at the view of the general population. Its hard to simulate finger prices & their acceptability if they have never experienced this themselves. It is interesting that this population had a higher acceptability for blood sampling & shows potentially how urine & other body fluids may be perceived as less acceptable- this warrants further exploration. Overall a well written piece that acknowledged its limitations in design & analysis, but start a good debate around sampling in this population.
--

REVIEWER	Reviewer name: Joyce Popoola Institution and Country: United Kingdom of Great Britain and Northern Ireland Competing interests: None
REVIEW RETURNED	02-Dec-2022

GENERAL COMMENTS	This article explores the views of children and young adults on micro-sampling and used a focus group and questionnaire to
--

	ascertain their perceptions and acceptability. The work answers important questions through in put of end users. There are limitations to the study for instance the wide variation in ages of participants cutting across different stages of life in terms of mental development. The number of participants is also relatively small. The study however was still able to draw out important information that has potential implication for the wider community around adopting micro-sampling for clinical and research purposes. There is also very limited data on the adolescent age involved in this study. Despite the limitations I see this study as having a message that is important to share with the wider research community and would therefore encourage its publication but with more emphasis on the limitations of the work in the discussion section.
--	---

VERSION 1 – AUTHOR RESPONSE

We would like to thank the editors and reviewers for their time and effort in reviewing our work and offering suggestions that have allowed us to improve our manuscript. All changes suggested

have been made and can be seen in the marked version as track changes. Our response to each

individual comment is detailed below in bold italic text. We hope these changes are adequate for

the reviewers and editors, please let us know if there are any other comments you have regarding

work to help us improve if required.

Editor in Chief Comments to Author:

Thank you for evaluating our work and offering suggestions to help us improve our manuscript.

We hope the amendments adequately resolve the queries raised.

Title amend to "Perceptions and acceptability of micro-sampling in children and young people; a

single centre survey"

We have changed title to the suggested version.

Abstract avoid the use of % for a sample of 13. Use actual numbers.

Thank you for the suggestion we have amended as such.

Abstract Conclusions add "in this small study" after "young people"

This has now been amended

What this study adds delete the 1st statement as it is methods

This has now been removed.

2nd statement amend to "The small group of children and young persons in this study were happy

with the collection of dried blood spots"

We have amended this statement to reflect the suggestion.

Text and Figures avoid the use of %

We have removed all instances of % in the manuscript other than one where we are refereeing to

data from a cited reference where only a % was provided.

Conclusions delete the two sentences stating your study is the first. Journal policy is for authors to

avoid describing their study as the first.

Apologies for this oversight, we have now deleted these two sentence

Do NOT overstate the findings from your small study or we will have to reject your paper

We have added a new section to the discussion focussing on the limitations of this work such as

the sample size and the fact the participants have an interest in medical research. We have also

added the the results of the abstract that only 10 of the 13 children responded to the questionnaire

to clarify as soon as possible in the manuscript both the number of children in the study overall

and the number who responded to the questionnaire.

Associate Editor

Comments to the Author:

Please ensure that the discussion covers the limitations (and their implications) that the reviewers

have pointed out.

Thank you for your comment, we have now added a section to the discussion titled study limitation which discuss the limitations of this study considering all of the comments by the 2 reviewers and editors.

Reviewer: 1

Dr. Helen Sammons, North Devon District Hospital, North Devon District Hospital

Comments to the Author

A well written piece that adds to our knowledge. A small descriptive group discussion that is documented to show children & young people's views on microsampling. The sample size is biased &

small (13 attended & 10 completed a non-validated survey). The paper acknowledges this noting it is

also an interested population who have a health care/research interest. This could be brought out

further in the abstract. It should also be highlighted further in the limitations the fact it was an online session with only one participant having hands on access to the kit.

It however gives the first evidence of this group's thoughts. It will hopefully be the start of further

consumer groups to look at the view of the general population. It's hard to simulate finger pricks &

their acceptability if they have never experienced this themselves.

It is interesting that this population had a higher acceptability for blood sampling & shows potentially how urine & other body fluids may be perceived as less acceptable- this warrants further

exploration.

Overall a well written piece that acknowledged its limitations in design & analysis, but start a good

debate around sampling in this population.

Thank you for your kind evaluation of our work, we acknowledge that there are weaknesses to our

study and we have now added a section to the end of the discussion detailing these to the reader.

Reviewer: 2

Joyce Popoola

Comments to the Author

This article explores the views of children and young adults on micro-sampling and used a focus

group and questionnaire to ascertain their perceptions and acceptability. The work answers important questions through input of end users.

There are limitations to the study for instance the wide variation in ages of participants cutting across different stages of life in terms of mental development. The number of participants is also

relatively small. The study however was still able to draw out important information that has potential implication for the wider community around adopting micro-sampling for clinical and research purposes. There is also very limited data on the adolescent age involved in this study.

Despite the limitations I see this study as having a message that is important to share with the wider

research community and would therefore encourage its publication but with more emphasis on the

limitations of the work in the discussion section.

We would like to thank the reviewer for their kind words about our work. It is true there are a number of limitations with this study and have now added the appropriate section at the end of the

discussion to acknowledge the impact these may have on our work. We hope that this section alleviates the problem raised by the reviewer.

VERSION 2 – REVIEW

REVIEWER	Reviewer name: Institution and Country: Competing interests:
REVIEW RETURNED	

GENERAL COMMENTS	
--

REVIEWER	Reviewer name: Institution and Country: Competing interests:
REVIEW RETURNED	

GENERAL COMMENTS	
--

REVIEWER	Reviewer name: Institution and Country: Competing interests:
REVIEW RETURNED	

GENERAL COMMENTS	
--

VERSION 2 – AUTHOR RESPONSE

VERSION 3 – REVIEW

REVIEWER	Reviewer name: Institution and Country: Competing interests:
REVIEW RETURNED	

GENERAL COMMENTS	
--

REVIEWER	Reviewer name: Institution and Country: Competing interests:
REVIEW RETURNED	

GENERAL COMMENTS	
--

REVIEWER	Reviewer name: Institution and Country: Competing interests:
REVIEW RETURNED	

GENERAL COMMENTS	
--

VERSION 3 – AUTHOR RESPONSE